# Factors associated with under-reporting of head and neck squamous cell carcinoma in cause-of-death records: A comparative study of two national databases in France from 2008 to 2012

Caroline Even[1], Luis Sagaon Teyssier[2,3], Yoann Pointreau[4], Stéphane Temam[1], Florence Huguet[5], Lionnel Geoffrois[6], Michaël Schwarzinger[3,¤a,¤b]*, on behalf of the EPICORL Study Group[¶]

1 Department of Surgical & Medical Head & Neck Oncology, Gustave Roussy Cancer Campus, Villejuif, France, 2 Aix Marseille Univ, INSERM, IRD, SESSTIM, Marseille, France, 3 Translational Health Economics Network (THEN), Paris, France, 4 Department of Radiation Oncology, ILC- Institut inter-régionaL de Cancérologie, Centre Jean Bernard-Clinique Victor Hugo, Le Mans, France, 5 Department of Radiation Oncology, Hôpital Tenon, AP-HP, Paris, France, 6 Department of Medical Oncology, Institut de cancérologie de Lorraine – Alexis Vautrin, Vandoeuvre Les Nancy, France

¤a Current address: Department of Methodology and Innovation in Prevention, Bordeaux University Hospital, Bordeaux, France
¤b Current address: Université de Bordeaux, Isped, Inserm UMR 1219-Bordeaux Population Health, Bordeaux, France
¶ Membership of the EPICORL Study Group is provided in the Acknowledgments.
* michael.schwarzinger@chu-bordeaux.fr

**Data Availability Statement:** French UCoD statistics are publicly available from the

## Abstract

### Objective

To date, no study has evaluated the detection rate of head and neck squamous cell carcinoma (HNSCC) in cause-of-death records in Europe. Our objectives were to compare the number of deaths attributable to HNSCC from two national databases in France and to identify factors associated with under-reporting of HNSCC in cause-of-death records.

### Methods

The national hospital discharge database and the national underlying cause-of-death records were compared for all HNSCC-attributable deaths in adult patients from 2008 to 2012 in France. Factors associated with under-reporting of HNSCC in cause-of-death records were assessed using multivariate Poisson regression.

### Results

A total of 41,503 in-hospital deaths were attributable to HNSCC as compared to 25,647 deaths reported in national UCoD records (a detection rate of 62%). Demographics at death were similar in both databases with respect to gender (83% men), age (54% premature deaths at 25–64 years), and geographic distribution. In multivariate Poisson regression, under-reporting of HNSCC in cause-of-death records significantly increased in 2012

Epidemiological Center on Medical Causes of Death (CépiDc-INSERM) at https://www.cepidc.inserm.fr/causes-medicales-de-deces/interroger-les-donnees-de-mortalite. De-identified data from the 2008-2013 French national hospital discharge (PMSI) database are available from the Agence Technique de l'Information sur l'Hospitalisation (ATIH) at https://www.atih.sante.fr/bases-de-donnees/commande-de-bases.

**Funding:** The EPICORL study was supported by a research grant from Merck Sharp & Dohme (MSD) France. Translational Health Economics (THEN) was responsible for data collection, preparation, and statistical analysis. THEN received payment from Merck Sharp & Dohme (MSD) France for these research activities. The funders had no role in study design, data collection and analysis, decision to publish, or preparation of the manuscript.

**Competing interests:** I have read the journal's policy and the authors of this manuscript have the following competing interests: Dr Caroline Even reports personal fees from Astra Zeneca, BMS, Innate Pharma, Merck and co, and Merck Serrono, outside and unrelated to the submitted work; Luis Sagaon-Teyssier was an employee of Translational Health Economics Network (THEN); Prof Françoise Huguet reports personal fees from BMS and Merck Serrono, outside and unrelated to the submitted work; Dr Michaël Schwarzinger was an employee of Translational Health Economics Network (THEN), Paris, France, that received a research grant from Merck Sharp & Dohme (MSD) France for the EPICORL study as well as research grants from Abbvie, Gilead and Novartis, outside and unrelated to the submitted work. Other authors have declared no conflicts of interest. This does not alter our adherence to PLOS ONE policies on sharing data and materials. There are no patents, products in development, or marketed products to declare.

compared to 2010 (+7%) and was independently associated with a primary HNSCC site other than the larynx, a former primary or second synchronous cancer other than HNSCC, distant metastasis, palliative care, and death in hospitals other than comprehensive cancer care centers. The main study results were robust in a sensitivity analysis which also took into account deaths outside hospital (overall, 51,129 HNSCC-attributable deaths; a detection rate of 50%). For the year 2012, the age-standardized mortality rate for HNSCC derived from underlying cause-of-death records was less than half that derived from hospital discharge summaries (14.7 compared to 34.1 per 100,000 for men and 2.7 compared to 6.2 per 100,000 for women).

## Conclusion

HNSCC is largely under-reported in cause-of-death records. This study documents the value of national hospital discharge databases as a complement to death certificates for ascertaining cancer deaths.

## Introduction

National underlying cause-of-death (UCoD) records are a cornerstone of epidemiological research, health policy planning, and evaluation [1]. UCoD is certified by a physician using the coding rules and procedures of the WHO International Classification of Diseases (ICD), which is currently available in its tenth revision (ICD-10) [2]. However, many studies have identified misclassification problems in UCoD records, due to incomplete or inaccurate death certification [3, 4]. This was particularly the case for older decedents [5–7] and for certain specific UCoDs [7–9]. Despite coding rules and procedures which give priority to cancer over comorbidities as the UCoD [2], cancer deaths also present misclassification problems [10–12]. In particular, three U.S. studies conducted on population-based cancer registries have all identified low detection rates (around 60%) for head and neck squamous cell carcinoma (HNSCC) in UCoD records [10, 11, 13].

To our knowledge, the detection rate for HNSCC in UCoD records has not been studied in Europe. In several European countries, national hospital discharge records may be used to ascertain cancer deaths [14–17]. For instance, all hospital discharge records in France document diagnoses using standardized ICD-10 coding. In a recent French study linking death certificates with the last hospital admission in 2008 and 2009, the primary discharge diagnosis was similar to the UCoD for 54% of cancer deaths [18]. However, previous studies have relied only on the discharge records of the last acute hospital admission before death and, for this reason, may not fully capture the "chain of events leading directly to death" [14–18].

The objective of this study was to estimate the number of deaths attributable to HNSCC in France in 2008–2012 from all hospital discharge records in acute and post-acute care. This number was compared to national UCoD records for HNSCC in 2008–2012. In addition, we assessed factors associated with under-reporting of HNSCC in national UCoD records.

## Materials and methods

### Study design

Firstly, we conducted a retrospective longitudinal analysis of the French national hospital discharge database (Programme de Médicalisation des Systèmes d'Information; PMSI 2008–

2013) to identify and characterize deaths attributable to HNSCC in 2008–2012. HNSCC was defined by the following ICD-10 codes: C00-C06; C09-C14; C30.0; C31; C32 [19]. The full coding dictionary used in the study is provided in S1 Table in S1 File. Secondly, we compared the number of HNSCC-attributable deaths identified in the national hospital discharge (PMSI) database to the number of deaths with HNSCC reported as the UCoD in national cause-of-death records, using the same definition of HNSCC, geographical area, and period. Finally, we assessed factors associated with under-reporting of HNSCC in national UCoD records, taking into account all patient characteristics available for analysis at the time of death.

## Data sources

The national hospital discharge (PMSI) database contains discharge records for all acute hospital admissions and post-acute care in the public and private sectors. All hospital discharge records for a given patient are linked by a unique patient identification number [20, 21]. National UCoD records were obtained from the national mortality database (Epidemiological Center on Medical Causes of Death, CépiDc-INSERM 2008–2012).

## Identification of HNSCC-attributable deaths in the national hospital discharge database

The study sample consisted of the 131,965 patients discharged with HNSCC in France between 2008 and 2012 (inclusive), of whom 46,463 (35.2%) died in hospital during the same period (S2 Table in S1 File). The set of rules to assess whether HNSCC was a probable/possible/unlikely cause of death was established by the medical experts of the study group in accordance with WHO coding rules and procedures prioritizing cancer as the UCoD over comorbidities [2, 22]. Accordingly, HNSCC was considered the probable cause of death in patients with distant metastases at initial treatment and in patients who relapsed at least six months after diagnosis. It was considered the possible cause of death in patients initially treated at a locally advanced stage. In all other situations, HNSCC was considered an unlikely cause of death.

## Characterization of HNSCC-attributable deaths in the national hospital discharge database

A number of characteristics that may potentially be associated with misclassification in UCoD records were documented. These include patient demographics (gender, age at death, and region of residence), primary HSNCC site [10, 11, 13], distant metastasis after HNSCC diagnosis [13], multiple primary cancers [22]), comorbidities [22] and end-of-life characteristics.

The primary HNSCC site was identified from the primary diagnosis recorded on the hospital discharge summary for first HNSCC surgery or panendoscopy. Primary HNSCC sites were grouped into nine categories, namely nasal cavity/paranasal sinuses, nasopharynx, lip, tongue, oral cavity, oropharynx, hypopharynx, larynx and ill-defined HNSCC [19].

Multiple primary cancers were classified in four categories based on their expected frequency in HNSCC patients. Second primary HNSCC sites were classified as lung cancer, esophageal cancer or other primary cancers [23, 24]. WHO coding rules and procedures prioritize the first primary cancer as the UCoD over a second primary cancer [2, 22]. For this reason, multiple primary cancers were classified according to their temporal relationship to the primary HNSCC site. Six categories of multiple primary cancers were defined, namely history of HNSCC documented at diagnosis of HNSCC, second synchronous HNSCC, second metachronous HNSCC, history of non-HNSCC documented at diagnosis of HNSCC, second synchronous non-HNSCC and second metachronous non-HNSCC.

WHO coding rules specify that HIV/AIDS should be considered as the UCoD in HNSCC patients [2, 22, 25]. However, other comorbidities are frequent competing causes of death in patients with HNSCC [26, 27]. The Charlson comorbidity index is widely used in HNSCC care [26, 28] and we documented each comorbidity of the index using a validated ICD-10 coding algorithm for the national hospital discharge (PMSI) database [29, 30]. In addition, we documented depression and suicide attempts, both of which are frequently reported in HNSCC patients [31].

Finally, with respect to end-of-life characteristics, we documented palliative care provided by non-cancer specialists and the place of death (comprehensive cancer care center, public teaching hospital, private clinic, or public local hospital). These are two factors which may be associated with a lower degree of cancer specialization of the certifying physician and thus potentially more misclassification in UCoD records.

### Identification of HNSCC-attributable deaths in national cause-of-death records

National UCoD records are available by ICD-10 code and stratified by year of death (from 2008 to 2012), gender, age at death (eight categories: 25–34; 35–44; 45–54; 55–64; 65–74; 75–84; 85–94; 95+) and region of residency (22 regions). We selected UCoD records using the same HNSCC definition as for the national hospital discharge (PMSI) database. These were grouped into the same nine categories of primary HNSCC sites. Because of small sample sizes at extreme ages and in some regions, age at death was grouped into six categories (25–44, 45–54, 55–64, 65–74, 75–84 and 85+ years) and regions into five categories (Greater Paris region, North-West, North-East, South-West and South-East).

### Study outcome

The primary outcome of the analysis was the number of HNSCC-attributable deaths identified in the two national databases. In the national hospital discharge (PMSI) database, attributable deaths were defined as HNSCC identified as a probable/possible cause of in-hospital death. In the National UCoD records, they were identified as deaths with the UCoD reported as HNSCC. This number was compared between the two databases, both overall and stratified by the variables common to both datasets (year of death, gender, age category, region of residence, primary HNSCC site).

### Statistical analysis

The number of deaths attributable to HNSCC was compared for each of the 2,700 profiles defined by all possible combinations of the variables common to both databases. We studied under-reporting of HNSCC in UCoD records by selecting all profiles for which HNSCC-attributable deaths identified in the national hospital discharge (PMSI) database exceeded those identified in national UCoD records. A multivariate Poisson regression model was conducted in order to identify independent factors associated with under-reporting. All variables were entered in the initial model, which was then iterated with stepwise selection of variables ($p<0.10$ for entry and $p<0.05$ for retention). Variables common to both databases were included in all selection models. For these variables, exponents of Poisson coefficients correspond to relative risks (RR) of under-reporting. An RR>1 indicates an increased risk of under-reporting for a given category as compared to the reference category, and an RR <1 indicating a decreased risk.

For variables only identifiable in the national hospital discharge (PMSI) database, the number of in-hospital deaths was counted for each profile and the counts log-transformed. Log-

transformation made it possible both to control data dispersion and to interpret Poisson coefficients as elasticities. Accordingly, an elasticity>0 (<0) indicates the percentage increase (decrease) in the number of records under-reported given an increase of 1% in the number of deaths for the variable.

Since only in-hospital deaths are recorded in the national hospital discharge (PMSI) database, we conducted a sensitivity analysis with overall deaths after imputing deaths outside hospital [32]. Imputation was based on the analysis of multiple determinants of death for all patients discharged with HNSCC in 2008 and for whom vital status was known at the end of 2008. A multivariate logistic regression was then used to estimate the probability of dying outside hospital for each patient lost to follow-up from 2008 to 2012 (further details on imputation method are provided in S1 Methods in S1 File).

From the overall number of HNSCC-attributable deaths in the most recent year (2012), we computed age-standardized mortality rates (ASMR) by gender and region after standardization to the new European Standard Population [33]. All statistical analyses were carried out using the R software for records (version 3.2.0; R Foundation for Statistical Computing, Vienna, Austria).

### Ethics and consent

The EPICORL (EPIdémiologie des Cancers ORL) study was approved by the French National Commission for Data Protection (CNIL DE-2015-025) who granted access to the national hospital discharge (PMSI) database for the years 2008 to 2013. The requirement for informed consent was waived because the study used de–identified data.

## Results

### Deaths attributable to HNSCC in the national hospital discharge database

Of the 46,463 patients discharged with HNSCC and dying in hospital in 2008–2012, 41,503 (89.3%) deaths were attributable to HNSCC as a probable (28,254 [68.1%]) or possible (13,249 [31.9%]) cause of death, and 4,960 (10.7%) deaths were not attributable to HNSCC (Table 1). These data were left-censored on January 1st 2008, when the national hospital discharge (PMSI) database first became available. For this reason, identification of relapses requiring at least six months of follow-up after diagnosis (a criterion for attribution of HNSCC as a probable cause of death) was incomplete for the year 2008. The proportion of the study sample fulfilling this criterion increased from 2008 (9.8%) up until 2010 (44.0%) and remained stable thereafter (51.1% in 2012) (S3 Table in S1 File). The proportion of deaths with HNSCC as a

**Table 1. Deaths attributable to HNSCC in the national hospital discharge database (2008–2012).**

| Stage at initial treatment | Relapse during follow-up | Deaths attributable to HNSCC | Main analysis (in-hospital deaths only) | Sensitivity analysis (overall deaths)[a] |
|---|---|---|---|---|
| Distant metastasis stage | – | Probable | 10,882 (23.4) | 13,199 (22.4) |
| Locally advanced stage | Yes | Probable | 14,218 (30.6) | 17,183 (29.2) |
| | No | Possible | 13,249 (28.5) | 16,933 (28.7) |
| Early stage | Yes | Probable | 3,154 (6.8) | 3,814 (6.5) |
| | No | Unlikely | 4,960 (10.7) | 7,801 (13.2) |
| Total deaths attributable to HNSCC | | | 41,503 (89.3) | 51,129 (86.8) |
| Total deaths | | | 46,463 (100) | 58,930 (100) |

[a] In-hospital deaths or deaths imputed outside hospital in patients lost to follow-up after hospital discharge.

probable cause thus also increased over time, from 51.1% in 2008 to 70.9% in 2010 and 74.4% in 2012 (S3 Table in S1 File).

In a sensitivity analysis which also took into account deaths outside hospital in the 131,965 patients discharged with HNSCC in 2008–2012, overall all-cause mortality was estimated to be 44.7% (58,930 deaths) in 2008–2012 (S1 Methods in S1 File). Of these, 51,129 deaths (86.8%) were attributable to HNSCC, either as a probable (34,196 cases; 66.9%) or possible (16,933 cases; 30.1%) cause of death.

## Comparison of HNSCC-attributable deaths between the national hospital discharge database and national UCoD records

In the national UCoD records, 25,647 deaths were attributable to HNSCC over the 2008–2012 period. Accordingly, the detection rate of HNSCC-attributable deaths (as identified in the national hospital discharge database) in national UCoD records was 62% in the main analysis (41,503 in-hospital deaths) and 50% in the sensitivity analysis (51,129 overall deaths). Although the number of HNSCC-attributable deaths identified in the national hospital discharge (PMSI) database increased from 2008 to 2012, the number of HNSCC in national UCoD records remained relatively stable. In consequence, the detection rate decreased from 70% in 2008 to 56% in 2012 in the main analysis (from 57% to 43% in the sensitivity analysis) (Fig 1).

Apart from year of death, four characteristics of HNSCC-attributable deaths could be compared between the two national databases (Table 2). Demographics at death were similar in both databases regarding gender (83% men), age (54% premature deaths at 25–64 years), and geographic distribution. However, higher proportions of primary larynx cancers (24% vs. 19%) and ill-defined HNSCC (12% vs. 5%) were documented in national UCoD records than in the national hospital discharge (PMSI) database.

Other characteristics of HNSCC-attributable deaths could only be identified in the national hospital discharge (PMSI) database. Of the patients who died, 13% were recorded with multiple primary HNSCC, 43% with a primary cancer other than HNSCC (16% lung cancer, 8% esophageal cancer, and 20% other cancer; 10% former primary cancer, 22% second synchronous cancer, and 11% second metachronous cancer), 47% with distant metastasis after

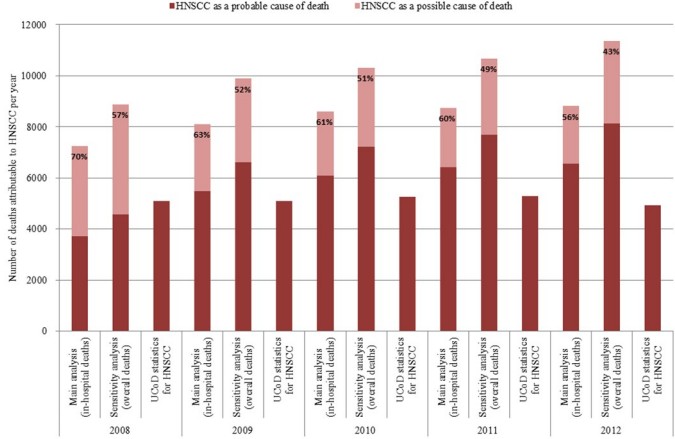

**Fig 1. Detection rate of HNSCC-attributable deaths in national UCoD records compared to the national hospital discharge database (2008–2012).** Percentages indicate the proportion of HNSCC-attributable deaths in the national hospital discharge (PMSI) database for which HNSCC is the certified underlying cause of death. Data are presented by year of death for both the main analysis and the sensitivity analysis.

**Table 2. Comparison of characteristics of HNSCC attributable-deaths according to the source of information (2008–2012).**

| Characteristics of HNSCC-attributable deaths | National hospital discharge (PMSI) database | | National UCoD records |
|---|---|---|---|
| | Main analysis (in-hospital deaths) | Sensitivity analysis (overall deaths)[a] | |
| | N = 41,503 | N = 51,129 | N = 25,647 |
| Year of death | | | |
| 2008 | 7,251 (17.5) | 8,887 (17.4) | 5,094 (19.9) |
| 2009 | 8,106 (19.5) | 9,886 (19.3) | 5,095 (19.9) |
| 2010 | 8,598 (20.7) | 10,309 (20.2) | 5,263 (20.5) |
| 2011 | 8,738 (21.1) | 10,673 (20.9) | 5,276 (20.6) |
| 2012 | 8,810 (21.2) | 11,374 (22.2) | 4,919 (19.2) |
| Men | 34,315 (82.7) | 42,133 (82.4) | 21,100 (82.3) |
| Age at death | | | |
| 25–44 | 934 (2.2) | 1,164 (2.3) | 502 (1.7) |
| 45–54 | 7,158 (17.3) | 8,670 (17.0) | 4,289 (16.7) |
| 55–64 | 14,296 (34.5) | 17,432 (34.1) | 8,572 (33.4) |
| 65–74 | 9,687 (23.3) | 11,935 (23.3) | 5,605 (21.9) |
| 75–84 | 7,196 (17.3) | 8,968 (17.5) | 4,682 (18.3) |
| ≥85 | 2,232 (5.4) | 2,960 (5.8) | 1,997 (7.8) |
| Region | | | |
| Greater Paris region | 6,449 (15.5) | 7,778 (15.2) | 3,657 (14.3) |
| North-West | 10,968 (26.4) | 13,381 (26.2) | 6,767 (26.4) |
| North-East | 11,237 (27.1) | 13,769 (26.9) | 7,033 (27.4) |
| South-West | 3,812 (9.2) | 4,819 (9.4) | 2,516 (9.8) |
| South-East | 9,037 (21.8) | 11,382 (22.3) | 5,674 (22.1) |
| Primary HNSCC site | | | |
| Nasal cavity/paranasal sinuses | 1,861 (4.5) | 2,277 (4.5) | 1,004 (3.9) |
| Nasopharynx | 1,212 (2.9) | 1,619 (3.2) | 612 (2.4) |
| Lip | 346 (0.8) | 468 (0.9) | 170 (0.7) |
| Tongue | 6,160 (14.8) | 7,558 (14.8) | 4,213 (16.4) |
| Oral cavity | 6,177 (14.9) | 7,439 (14.5) | 2,977 (11.6) |
| Oropharynx | 8,771 (21.1) | 10,793 (21.1) | 4,363 (17.0) |
| Hypopharynx | 6,935 (16.7) | 8,435 (16.5) | 3,217 (12.5) |
| Larynx | 7,864 (18.9) | 9,860 (19.3) | 6,064 (23.6) |
| Ill-defined HNSCC | 2,177 (5.2) | 2,680 (5.2) | 3,027 (11.8) |

[a] In-hospital deaths or deaths imputed outside hospital in patients lost to follow-up after hospital discharge.

Note: Results are presented as n (%).

HNSCC diagnosis and 64% with comorbidities, including 0.5% with HIV/AIDS and 11% with depression (S4 Table in S1 File). Two-thirds of patients who died received palliative care and less than 6% died in a comprehensive cancer care center. Similar proportions were found in the sensitivity analysis which also took into account deaths outside hospital (S4 Table in S1 File).

## Independent factors associated with under-reporting of HNSCC in UCoD records

National UCoD records for HNSCC were available for 2,700 profiles at death, defined by all possible combinations of year of death, gender, age group, region and primary HNSCC site. For 1,790 (66.3%) profiles, the number of HNSCC-attributable deaths at hospital was under-

reported in UCoD records. These profiles corresponded to 37,789 individual deaths, of which 17,474 were not identified as HNSCC-attributable in the UCoD records. Independent factors associated with under-reporting of these HNSCC-attributable deaths were evaluated in multi-variate Poisson regression.

Under-reporting was significantly higher in 2012 compared to 2010 (+7%) and was independently associated with intermediate ages (45–74 years), a primary HNSCC site other than the larynx, a former primary or second synchronous cancer other than HNSCC, any record of distant metastasis after HNSCC diagnosis, HIV/AIDS, depression, palliative care, and death in hospitals other than a comprehensive cancer care center (Table 3). In contrast, primary esophageal cancer was independently associated with less under-reporting compared to cancer of the larynx.

In the sensitivity analysis, adding HNSCC-attributable deaths outside hospital was associated with more under-reporting of HNSCC-attributable deaths in the UCoD records. This was the case for 2,054 (76.1%) of the 2,700 profiles, corresponding to 48,688 individual HNSCC-attributable deaths of which 26,489 were not identified as HNSCC-attributable in the UCoD records.

In the sensitivity analysis, the same factors associated with under-reporting were identified by multivariate Poisson regression as in the main analysis (Table 3). However, the strengths of these associations were generally lower than in the main analysis, except for the year of death (+10% in 2012 as compared to 2010) and gender (+7% in men as compared to women). In addition, under-reporting was independently associated with dying at home, which, together with public local hospitals, was the place of death associated with the highest rate of under-reporting.

## Age-standardized mortality rate of HNSCC in France in 2012

For the year 2012, the age-standardized mortality rate for HNSCC derived from underlying cause-of-death records was less than half that derived from hospital discharge summaries (14.7 compared to 34.1 per 100,000 for men and 2.7 compared to 6.2 per 100,000 for women) (Figs 2–5).

## Discussion

This study showed significant under-reporting of deaths attributable to HNSCC in national UCoD records as compared to the national hospital discharge (PMSI) database in France. Under-reporting was multi-factorial and has increased over recent years. For the year 2012, the age-standardized mortality rate of HNSCC derived from national UCoD records was less than half of the corresponding figure derived from the national hospital discharge database.

Misclassification problems in national UCoD records have been identified in many studies [3, 4, 6, 8, 9], including studies conducted in the French healthcare setting [5, 7, 12, 34]. The most commonly cited reasons for misclassification include physician inexperience in death certification and lack of appropriate training. In addition, retrospective determination of the "chain of events leading directly to death" may be problematic if the certifying physician was not involved in the care of the patient or has only limited access to the patient's medical records. This situation seems particularly critical for HNSCC-attributable deaths, since exposure to tobacco and alcohol are major risk factors for HNSCC and may be responsible for multiple other causes of death declared on death certificates [35–37].

Previous studies on misclassification of HNSCC in UCoD records have all been conducted using U.S. population-based cancer registries. These have described a detection rate for HNSCC-attributable deaths in UCoD records of around 60% [10, 11, 13]. Our results are

**Table 3. Variables associated with under-reporting of HNSCC in national UCoD records in multivariate Poisson regression (2008–2012).**

| Characteristics at death | Main analysis (N = 17,474 in-hospital deaths)[a] | | | Sensitivity analysis (N = 26,489 overall deaths)[b] |
|---|---|---|---|---|
| **Characteristics available in both databases** | **Relative risk (95% CI)** | **P-value** | **Relative risk (95% CI)** | **P-value** |
| Year of death (ref. 2010) | | | | |
| 2008 | 0.94 (0.88 to 1.01) | 0.079 | 0.91 (0.86 to 0.96) | <0.001 |
| 2009 | 1.00 (0.95 to 1.05) | 0.95 | 0.98 (0.94 to 1.02) | 0.38 |
| 2011 | 0.99 (0.95 to 1.04) | 0.81 | 1.00 (0.96 to 1.04) | 0.95 |
| 2012 | 1.07 (1.02 to 1.12) | 0.008 | 1.10 (1.05 to 1.14) | <0.001 |
| Men (ref. women) | 1.00 (0.94 to 1.07) | 0.91 | 1.07 (1.01 to 1.13) | 0.015 |
| Age at death (ref. 55–64) | | | | |
| 25–44 | 0.93 (0.82 to 1.06) | 0.28 | 0.81 (0.73 to 0.90) | <0.001 |
| 45–54 | 1.11 (1.05 to 1.17) | <0.001 | 1.05 (1.01 to 1.10) | 0.025 |
| 65–74 | 1.10 (1.04 to 1.15) | <0.001 | 1.07 (1.03 to 1.11) | <0.001 |
| 75–84 | 1.04 (0.98 to 1.11) | 0.22 | 1.02 (0.97 to 1.08) | 0.42 |
| ≥85 | 0.84 (0.75 to 0.94) | 0.003 | 0.81 (0.74 to 0.88) | <0.001 |
| Region (ref. Greater Paris region) | | | | |
| North-West | 0.95 (0.90 to 1.01) | 0.11 | 0.97 (0.93 to 1.02) | 0.29 |
| North-East | 0.94 (0.89 to 1.00) | 0.059 | 0.96 (0.91 to 1.01) | 0.083 |
| South-West | 1.00 (0.93 to 1.08) | 0.98 | 0.97 (0.91 to 1.03) | 0.26 |
| South-East | 0.95 (0.90 to 1.00) | 0.060 | 0.99 (0.94 to 1.03) | 0.55 |
| Primary HNSCC site (ref. Larynx) | | | | |
| Nasal cavity/paranasal sinuses | 1.71 (1.55 to 1.89) | <0.001 | 1.30 (1.21 to 1.41) | <0.001 |
| Nasopharynx | 1.76 (1.58 to 1.97) | <0.001 | 1.37 (1.25 to 1.49) | <0.001 |
| Lip | 1.50 (1.27 to 1.76) | <0.001 | 1.06 (0.93 to 1.21) | 0.38 |
| Tongue | 1.42 (1.33 to 1.52) | <0.001 | 1.23 (1.17 to 1.30) | <0.001 |
| Oral cavity | 2.04 (1.92 to 2.17) | <0.001 | 1.60 (1.53 to 1.68) | <0.001 |
| Oropharynx | 1.89 (1.79 to 2.01) | <0.001 | 1.54 (1.47 to 1.60) | <0.001 |
| Hypopharynx | 2.03 (1.91 to 2.16) | <0.001 | 1.59 (1.52 to 1.67) | <0.001 |
| Ill-defined HNSCC | 1.27 (1.09 to 1.47) | 0.002 | 0.93 (0.84 to 1.04) | 0.22 |
| **Characteristics only identified in the national hospital discharge database** | **Elasticity (95% CI)** | **P-value** | **Elasticity (95% CI)** | **P-value** |
| Cancer characteristics | | | | |
| Any esophageal cancer | -0.06 (-0.12 to -0.00) | 0.041 | -0.06 (-0.10 to -0.01) | 0.013 |
| Former primary cancer other than HNSCC | 0.06 (0.01 to 0.10) | 0.023 | 0.02 (-0.02 to 0.06) | 0.25 |
| Second synchronous cancer other than HNSCC | 0.10 (0.03 to 0.17) | 0.004 | 0.04 (-0.01 to 0.10) | 0.15 |
| Second metachronous cancer other than HNSCC | 0.02 (-0.03 to 0.06) | 0.49 | -0.01 (-0.05 to 0.03) | 0.68 |
| Any distant metastasis after HNSCC diagnosis | 0.15 (0.06 to 0.24) | <0.001 | 0.16 (0.08 to 0.23) | <0.001 |
| Comorbidities | | | | |
| HIV/AIDS | 0.10 (0.01 to 0.19) | 0.037 | 0.05 (-0.01 to 0.12) | 0.11 |
| Depression | 0.06 (0.01 to 0.11) | 0.014 | 0.04 (0.00 to 0.08) | 0.038 |
| End-of-life characteristics | | | | |
| Palliative care | 0.17 (0.05 to 0.29) | 0.005 | 0.21 (0.12 to 0.30) | <0.001 |
| Place of death: | | | | |
| Comprehensive cancer care center | 0.03 (-0.01 to 0.08) | 0.15 | 0.01 (-0.03 to 0.05) | 0.65 |
| Public teaching hospital | 0.20 (0.14 to 0.25) | <0.001 | 0.12 (0.07 to 0.16) | <0.001 |
| Private clinic | 0.17 (0.11 to 0.24) | <0.001 | 0.14 (0.09 to 0.18) | <0.001 |
| Public local hospital | 0.36 (0.27 to 0.46) | <0.001 | 0.28 (0.21 to 0.36) | <0.001 |
| Home (death imputed outside hospital) | — | | 0.25 (0.20 to 0.29) | <0.001 |

[a] Overdispersion test: z = -1.18, p-value = 0.88.

[b] In-hospital deaths or deaths imputed outside hospital in patients lost to follow-up after hospital discharge. Overdispersion test: z = 4.62, p-value = 1.

A relative risk >1 corresponds to an increased risk of under-reporting in a given category as compared to the reference category; a relative risk <1 corresponds to a decreased risk. An elasticity>0 indicates the percentage increase in the number of records under-reported given a 1% increase in the number of deaths for the variable; an elasticity<0 indicates the percentage decrease in the number of records under-reported.

UCoD: underlying cause-of-death; HNSCC: head and neck squamous cell carcinoma; HIV: Human Immunodeficiency Virus infection; AIDS: Acquired Immune Deficiency Syndrome; ref.: reference; CI 95%: confidence intervals at 95%.

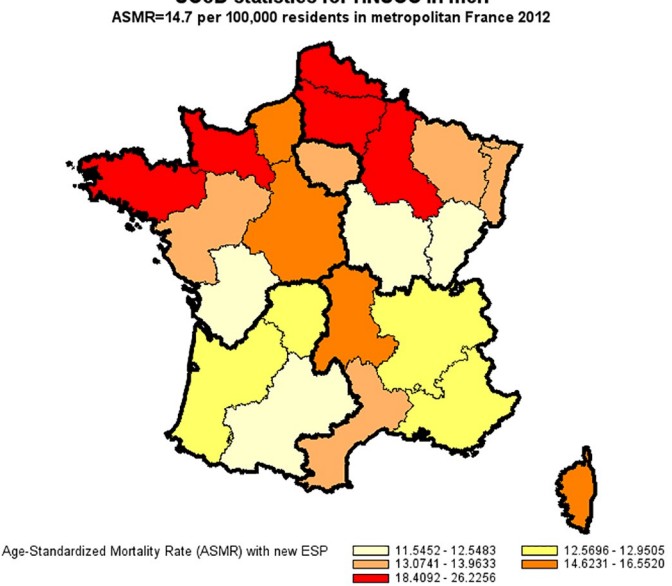

**Fig 2. Age-standardized mortality rate of HNSCC in men derived from UCoD records (France, 2012).** Age-Standardized Mortality Rates (ASMR) were standardized to the new European Standard Population (ESP 2011–2030).

consistent with these findings, with a detection rate of HNSCC-attributable deaths at hospital in French UCoD records of 62% when considering only in-hospital death records (main analysis) and even lower (50%) when considering overall deaths in a sensitivity analysis. We also found that primary larynx cancer (19%) was significantly more likely to be reported in UCoD records compared to all other primary HNSCC sites [10, 11]. We also found that any distant

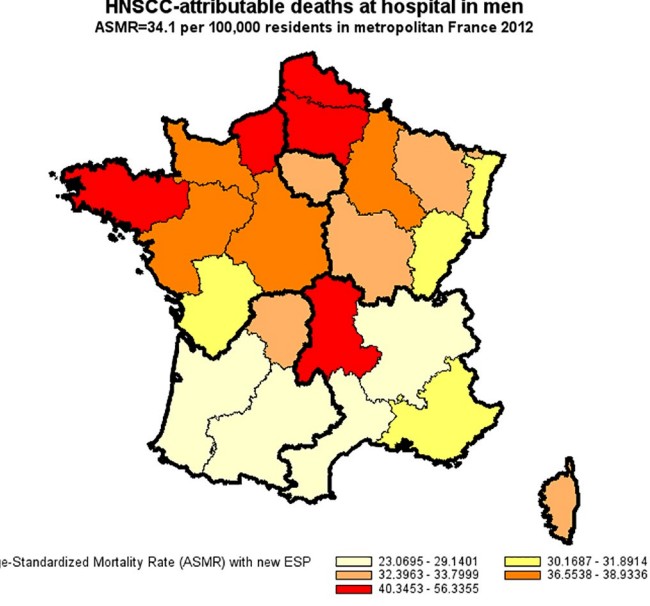

**Fig 3. Age-standardized mortality rate of HNSCC in men derived from the national hospital discharge database (France, 2012).** Age-Standardized Mortality Rates (ASMR) were standardized to the new European Standard Population (ESP 2011–2030).

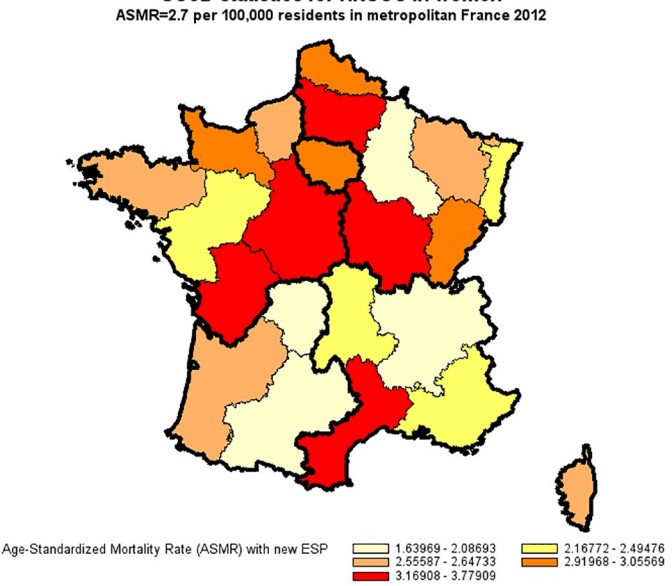

**Fig 4. Age-standardized mortality rate of HNSCC in women derived from UCoD records (France, 2012).** Age-Standardized Mortality Rates (ASMR) were standardized to the new European Standard Population (ESP 2011–2030).

metastasis recorded after HNSCC diagnosis (47%) was associated with an increased risk of HNSCC being under-reported in UCoD records [13]. These findings suggest that certifying physicians may be less attentive about specifying primary HNSCC sites associated with poor prognosis in death certificates [19].

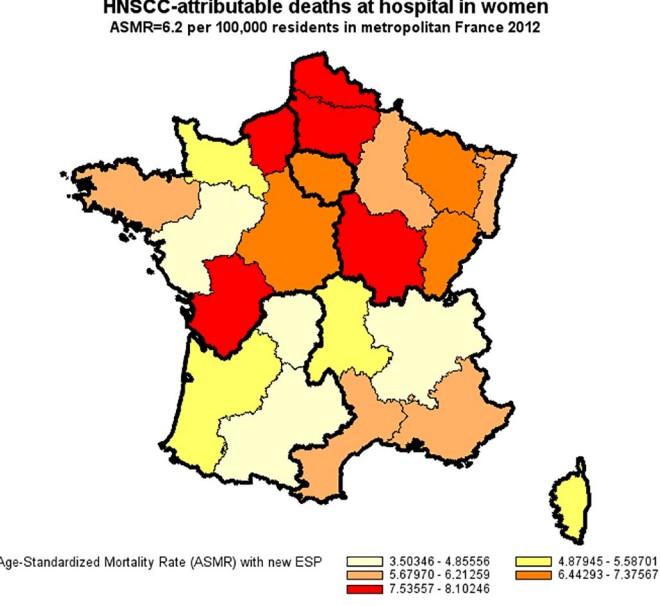

**Fig 5. Age-standardized mortality rate of HNSCC in women derived from the national hospital discharge database (France, 2012).** Age-Standardized Mortality Rates (ASMR) were standardized to the new European Standard Population (ESP 2011–2030).

This study using data from the national hospital discharge (PMSI) database provides new information on the frequency and nature of competing causes of death in patients with HNSCC. Previous studies have been conducted in cancer registries which do not usually record comorbidities and have excluded multiple primary cancers from the analysis [10, 11, 13]. For the deaths considered to be HNSCC-attributable from the national hospital discharge (PMSI) database, the presence of competing causes of death was the rule rather than the exception: 64% of HNSCC-attributable deaths were associated with comorbidities, 13% with multiple HNSCC and 43% with a primary cancer other than HNSCC. WHO coding rules and procedures prioritize cancer as the UCoD over comorbidities, with the exception of HIV/AIDS [2, 22]. Consistent with these recommendations, we did not find that comorbidities were associated with under-reporting of HNSCC in UCoD records (except for HIV/AIDS and depression).

Regarding multiple primary cancers, WHO coding rules and procedures prioritize reporting of the first primary cancer as the UCoD over a second primary cancer [2, 22]. We found that another primary cancer treated before HNSCC (10%) was significantly associated with under-reporting of HNSCC in the UCoD records. Nonetheless, primary esophageal cancer (8%) was significantly associated with better reporting of HNSCC. This may suggest that certifying physicians are more careful about specifying the primary HNSCC site as the UCoD when multiple cancers of the same body system are present, and supports recoding practices considering the first primary HNSCC site as the UCoD rather than esophageal cancer [22]. We found that a second synchronous cancer other than HNSCC was associated with under-reporting of HNSCC in UCoD records. Since lung cancer was the most frequent second primary cancer (16.1%), and may arise from metastatic spread of HNSCC [2], our findings suggest that certifying physicians usually simplify the "chain of events leading directly to death" by reporting lung cancer as the UCoD rather than the primary HNSCC site.

Finally, our findings suggest that the cancer specialization of the certifying physician is important for accurately identifying a primary HNSCC site as the UCoD. Of 41,503 individuals dying in hospital in 2008–2012, 69% received palliative care provided by non-cancer specialists and 56% died in public local hospitals where cancer specialists are under-represented. In agreement with a previous study on the misclassification of uterine cancers in French UCoD records [38], we found that both markers of a lower cancer specialization of the certifying physician were strongly associated with under-reporting of HNSCC in UCoD records. Similarly, death certification at home is generally not performed by a cancer specialist in France and we found that death certification outside hospital (19% of overall deaths in the sensitivity analysis) was associated with a similar degree of under-reporting.

The study has a number of strengths and limitations, which are frequently interrelated. An important strength is the fact that the study is based on a nationwide sample of all patients discharged with HNSCC, for whom a full medical history could be reconstituted from all hospital discharge records from January 1st, 2008, until death in 2008–2012. The use of an algorithm to identify all HNSCC-attributable deaths in the national hospital discharge (PMSI) database avoids problems of subjectivity such as those associated with the judgment of the certifying physician in the UCoD records. On the other hand, the accuracy of the algorithm is critical for the robustness of the findings. The performance of the algorithm had not been externally validated prior to the study. Nonetheless, we found that HNSCC was identified as a probable/possible cause of death in more than 85% of cases within a short interval after diagnosis of HNSCC. This is consistent with data from French cancer registries indicating that HNSCC was the initial cause of death in 93% of deaths recorded within five years after diagnosis [39].

The detection of HNSCC-attributable deaths by the algorithm was logically sensitive to the length of follow-up at death due to left-censoring of the national hospital discharge (PMSI)

database on January 1ˢᵗ, 2008. Overall, we found that identification of a relapse in the follow-up was the main criterion by which HNSCC was determined as the probable cause of death (S3 Table in S1 File). In consequence, HNSCC was identified as the probable cause of death in 51% of HNSCC-attributable deaths in 2008 (all decedents had less than one year of follow-up) as compared to 74% in 2012 (decedents may have up to five years of follow-up). In this regard, our study results for the year 2012 are the most reliable and suggest that the detection rate of HNSCC in UCoD records may be as low as 43% (Fig 1).

Since follow-up was right-censored at last hospital discharge in 2008–2012, we conducted a sensitivity analysis to assess death outside hospital in patients who were lost to follow-up after hospital discharge (S1 Methods in S1 File). Of 51,129 HNSCC-attributable deaths estimated overall, 41,503 (81.2%) occurred in hospital, which is consistent with the proportion (73%) of cancer deaths in death certificates in which the hospital is given as the place of death [40]. These limitations due to censoring in the national hospital discharge (PMSI) database may be overcome in the future, as a result of ongoing efforts to facilitate broader access to the last ten years of the national hospital discharge database with information on vital status on December 31ˢᵗ of the last study year [41]. In addition, possible linkage of the national hospital discharge (PMSI) database to the national database of multiple causes of death would provide more specific information on whether HNSCC was simply not mentioned on the death certificate [18] or, as may be more likely, misclassified as an associated cause of death rather than the UCoD [12].

## Conclusions

HNSCC is largely under-reported in cause-of-death records in France. This study documents the value of national hospital discharge databases as a complement to death certificates for ascertaining cancer deaths in general and HNSCC-attributable deaths in particular.

## Supporting information

**S1 File.**
(DOCX)

## Acknowledgments

The EPICORL (EPIdémiologie des Cancers ORL) Study Group includes: Sylvain Baillot, MSc, Translational Health Economics Network (THEN), Paris, France; Mélina Bec, MSc, Health Economics & Outcomes Research department, MSD France; Lynda Benmahammed, MD, Medical Advisor Oncology, MSD France; Caroline Even, MD, PhD, Department of Head & Neck Surgical & Medical Oncology, Institut de cancérologie Gustave Roussy, Villejuif, France; Lionnel Geoffrois, MD, PhD, Department of Medical Oncology, Institut de cancérologie de Lorraine–Alexis Vautrin, Vandoeuvre Les Nancy, France; Florence Huguet, MD, PhD, Department of Radiation Oncology, Hôpital Tenon, AP-HP, France; Béatrice Le Vu, MD, MSc, Stratégie et Gestion Hospitalière, UNICANCER Fédération Nationale des Centres de Lutte Contre le Cancer, Paris, France & Translational Health Economics Network (THEN), Paris, France; Laurie Lévy-Bachelot, PhD, Health Economics & Outcomes Research department, MSD France; Stéphane Luchini, PhD, CNRS, GREQAM-IDEP, Marseille, France & Translational Health Economics Network (THEN), Paris, France; Yoann Pointreau, MD, PhD, Department of Radiation Oncology, ILC- Institut inter-régionaL de Cancérologie, Centre Jean Bernard-Clinique Victor Hugo, Le Mans, France; Camille Robert, PharmD, Health Economics & Outcomes Research department, MSD France; Luis Sagaon Teyssier, PhD,

AMU/Inserm/IRD, UMR 912, Marseille, France & Translational Health Economics Network (THEN), Paris, France; Antoine Schernberg, MD, MPH, Department of Radiation Oncology, Hôpital Tenon, AP-HP, Paris, France; Michaël Schwarzinger, MD, PhD, Translational Health Economics Network (THEN), Paris, France; Stéphane Temam, MD, PhD, Department of Head & Neck Surgical & Medical Oncology, Institut de cancérologie Gustave Roussy, Villejuif, France.

## Author Contributions

**Conceptualization:** Michaël Schwarzinger.

**Data curation:** Michaël Schwarzinger.

**Formal analysis:** Luis Sagaon Teyssier, Michaël Schwarzinger.

**Funding acquisition:** Michaël Schwarzinger.

**Investigation:** Caroline Even, Yoann Pointreau, Stéphane Temam, Florence Huguet, Lionnel Geoffrois.

**Methodology:** Luis Sagaon Teyssier, Michaël Schwarzinger.

**Project administration:** Michaël Schwarzinger.

**Supervision:** Michaël Schwarzinger.

**Validation:** Caroline Even, Yoann Pointreau, Stéphane Temam, Florence Huguet, Lionnel Geoffrois, Michaël Schwarzinger.

**Writing – original draft:** Michaël Schwarzinger.

**Writing – review & editing:** Caroline Even, Luis Sagaon Teyssier, Yoann Pointreau, Stéphane Temam, Florence Huguet, Lionnel Geoffrois, Michaël Schwarzinger.

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
