## [Decision Letter · Decision Letter 0]

15 Sep 2020

PONE-D-20-25023

Underlying cause-of-death statistics underestimate the burden of head and neck squamous cell carcinoma: a nationwide hospital study in France, 2008-2012

PLOS ONE

Dear Dr. Schwarzinger,

Thank you for submitting your manuscript to PLOS ONE. After careful consideration, we feel that it has merit but does not fully meet PLOS ONE’s publication criteria as it currently stands. Therefore, we invite you to submit a revised version of the manuscript that addresses the points raised during the review process.

A **rebuttal letter** that responds to **EACH** point raised by the academic editor and reviewer(s). You should upload this letter as a separate file labeled 'Response to Reviewers'.A **marked-up copy** of your manuscript that highlights changes made to the original version. You should upload this as a separate file labeled 'Revised Manuscript with Track Changes'.An **unmarked version** of your revised paper without tracked changes. You should upload this as a separate file labeled 'Manuscript'.

We look forward to receiving your revised manuscript.

Kind regards,

Brecht Devleesschauwer

Academic Editor

PLOS ONE

Journal Requirements:

2. Thank you for including your competing interests statement; "I have read the journal's policy and the authors of this manuscript have the following competing interests: Dr Caroline Even reports personal fees from Astra Zeneca, BMS, Innate Pharma, Merck and co, and Merck Serrono, outside and unrelated to the submitted work; Luis Sagaon-Teyssier was an employee of Translational Health Economics Network (THEN); Prof Françoise Huguet reports personal fees from BMS and Merck Serrono, outside and unrelated to the submitted work; Dr Michaël Schwarzinger is employee of Translational Health Economics Network (THEN), Paris, France, that received research grants from Merck Sharp & Dohme (MSD) France as well as Abbvie, Gilead and Novartis, outside and unrelated to the submitted work. Other authors have declared no conflicts of interest."

We note that you received funding from a commercial source:Astra Zeneca, BMS, Innate Pharma, Merck and co, and Merck Serrono, Abbvie, Gilead and Novartis

Additional Editor Comments (if provided):

In your revision note, please include EACH of the reviewer comments, provide your reply, and when relevant, include the modified/new text (or motivate why you decided not to modify the text). Note that failure to do so may result in a rejection of the manuscript.

Reviewers' comments:

Reviewer's Responses to Questions

**Comments to the Author**

1. Is the manuscript technically sound, and do the data support the conclusions?

Reviewer #1: Yes

Reviewer #2: Yes

2. Has the statistical analysis been performed appropriately and rigorously? 

Reviewer #1: Yes

Reviewer #2: Yes

3. Have the authors made all data underlying the findings in their manuscript fully available?

Reviewer #1: Yes

Reviewer #2: Yes

4. Is the manuscript presented in an intelligible fashion and written in standard English?

Reviewer #1: No

Reviewer #2: Yes

5. Review Comments to the Author

Reviewer #1: August 28, 2020

Manuscript Number: PONE-D-20-25023

“Underlying cause-of-death statistics underestimate the burden of head and neck squamous cell carcinoma: a nationwide hospital study in France, 2008 - 2012”

Dear Dr Brecht Devleesschauwer,

I deeply appreciate your kind consideration of me to review this manuscript. I reviewed it with great interest.

Authors have tried to highlight the underreporting of causes of death of head and neck squamous cell carcinoma (HNSCC) by comparing two national databases and identified some potential factors associated with that. The approach used in this study is interesting. The evidence of this study is of great importance to improve the coding of related deaths that would help to calculate the burden of disease estimates correctly.

The authors did not actually estimate the burden of death statistics of HNSCC but highlighted the under-reporting of death statistics. Therefore, I propose to revise the title of the manuscript such as “Factors associated with under-reporting of death statistics of HNSCC: A comparative study of two national databases in France from 2008 to 2012.”

The authors have performed a multivariate Poisson regression analysis to identify the factors associated with under-reporting of HNSCC death statistics. Overall analysis and the results are robust. However, this manuscript needs to improve the quality of writing throughout the manuscript and the structure of methods, results and discussion sections. Most of the sentences are too long and should split in short to make it clear and more comprehensible. Therefore, the authors should ask a native English speaker for its proofreading. Therefore, I proposed “major revisions” to improve the quality of writing of this manuscript. Please find my comments below this letter.

Sincerely,

Romana Haneef

Reviewer’s comments to authors

Formatting

Page numbers and line numbers are missing. The spacing between the references numbers is not appropriate. Please update the manuscript according to the guidelines of the journal.

Title:

The authors did not estimate the burden of death statistics of HNSCC but highlight the under-reporting of death statistics. Therefore, I propose to revise the title of the manuscript such as “Factors associated with under-reporting of death statistics of HNSCC: A comparative study of two national databases in France from 2008 to 2012.”

Abstract

Please add the following information:

• Objective: The authors can update and add another objective as follows: “Our objectives were to compare the reporting of death attributable to HNSCC from two national databases in France and to identify the underlying factors associated with under-reporting.”

• Methods: Authors used the word “The determinants of missing HNSCC…, it is better to replace it with under-reporting.

• Results: Please revise the sentence structures for the results section.

• Conclusions: Please revise the first sentence according to the updated title.

Introduction

• In the first paragraph, last sentence “to our knowledge […]”, authors used exposure to tobacco and alcohol account for larger population attributable fractions of HNSCC but did not take into account these factors in their analysis. It is unclear to me mention it here. I propose to remove it and rephrase it.

• Please replace the word “combining” with “linking” individual records […] at the beginning of the second paragraph.

• Please revise the text to improve the level of writing and avoid using words like instead, around, possibly.

Materials and methods

• Please add as a sub-heading the study design (part of the text from sub-section of the definition of HNSCC attributable deaths at hospital) and study population (inclusion and exclusion criteria from S2 table and cite S2 table under this section) just before the data sources.

• Please also add the definition of “unlikely cause of death” at the end of the sub-section of the definition of HNSCC attributable deaths at hospital).

• Please revise the text of the last paragraph of the sub-section of “characteristics of HNSCC attributable deaths at hospital” and mention the “end of life characteristics” as 1. palliative care and 2. place of death (under this characteristic, authors can report deaths recorded at different places). This description should correspond to the table, which is not numbered (I suppose, table 4).

• Please specify death at six age groups in the main text under the section of UCoD statistics for HNSCC.

• Please add the sub-section of study outcomes before the statistical analysis (the first paragraph of statistical analysis could be moved to the study outcome section).

• Please revise the text to improve the level of writing

Results

1. The authors can start the result section by reporting the “General characteristics of deaths attributable to HNSCC according to two national databases in France” as table 1 rather than table 2.

a. Please add the word “National”UCoD statistics in table 1 (currently table 2)

2. The second sub-section of results could be “Death attributable to HNSCC at hospital” as table 2.

a. Under this section, please also report the “unlikely death attributable to HNSCC” in the main text.

b. Authors can report the “relapse in the follow-up overall from 2008 to 2010 from S3table to the main text and in table 2 as well.

c. Citation of S2 table is not appropriate in this section.

d. Please remove the word “definition” from the legend of table 2 (currently table 1) and start with “Deaths attributable […]”.

e. In figure 1, please report the respective percentages of probable and possible causes of death attributable to HNSCC. Currently, it is unclear, for example in 2008, under in-hospital death, 70% corresponds to both causes of death (probable and possible)? Is this figure based on the data from S3 table?

3. The last paragraph of sub-section “Comparison of HNSCC […]”, please replace the word “multiple primary HNSCC” with “decedent”, which does not correspond to the table S4table.

a. Please update the title as follows: “Comparison of HNSCC […] with UCoD statistics”.

4. The last sub-section of the results is unclear with long sentences. I propose to revise it and split into short sentences to make it more comprehensible.

a. Under this sub-section, a table is reported and not numbered. I guess it should be table 4. Please add the table number and heading/legend of this table.

Discussion

• The authors should revise the overall text of the discussion section without repeating the same results, avoid long sentences, report strengths and limitations in a more clear text.

• This section could be started such as “The result of this study showed the underreporting of death attributable to HNSCC in national UCoD as compared to national hospital discharge database.”

• Avoid using words such as massive, unexpectedly, rather, indeed, altogether, etc.

• Authors mentioned again the same phrase as reported in the introduction section “To our knowledge […]”, it is nuclear. Please explain the reason to repeat this sentence.

Conclusions

Please update the conclusions section and do not report the results in this section.

Reviewer #2: Overall the paper is well written and the analyses presented is clear and strongly support the conclusions reached. Some minor comments to improve the paper are outlined below:

-The paper doesn't make it clear whether deaths with an associated cause of death recorded as HNSCC have been looked at. Analyses of Australian deaths data suggests that HNSCC deaths would increase by around one-third (33%) if associated causes of death are included. Can this be looked at in the French mortality data?. If not, or if no deaths are recorded with HNSCC as an associated cause, then suggest making this more explicit in the paper.

- The statistics reported in the paper to do include a variance measure (ie confidence intervals). Is it possible to include these, or if not, provide a reason why these have not been included or are deemed not to be needed for the analyses presented.

- Paper could benefit from a gramma check to fix minor grammatical errors eg in the abstract results - 'associated to' should be 'associated with'

6. PLOS authors have the option to publish the peer review history of their article (what does this mean?). If published, this will include your full peer review and any attached files.

Reviewer #1: **Yes: **Romana Haneef

Reviewer #2: **Yes: **Ms Michelle Gourley

---

## [Author Response · Author response to Decision Letter 0]

17 Nov 2020

Reviewer #1 Romana Haneef

Dear Dr Brecht Devleesschauwer,

I deeply appreciate your kind consideration of me to review this manuscript. I reviewed it with great interest.

Authors have tried to highlight the underreporting of causes of death of head and neck squamous cell carcinoma (HNSCC) by comparing two national databases and identified some potential factors associated with that. The approach used in this study is interesting. The evidence of this study is of great importance to improve the coding of related deaths that would help to calculate the burden of disease estimates correctly.

The authors did not actually estimate the burden of death statistics of HNSCC but highlighted the under-reporting of death statistics. Therefore, I propose to revise the title of the manuscript such as “Factors associated with under-reporting of death statistics of HNSCC: A comparative study of two national databases in France from 2008 to 2012.”

The authors have performed a multivariate Poisson regression analysis to identify the factors associated with under-reporting of HNSCC death statistics. Overall analysis and the results are robust. However, this manuscript needs to improve the quality of writing throughout the manuscript and the structure of methods, results and discussion sections. Most of the sentences are too long and should split in short to make it clear and more comprehensible. Therefore, the authors should ask a native English speaker for its proofreading. Therefore, I proposed “major revisions” to improve the quality of writing of this manuscript. Please find my comments below this letter.

Thank you for your positive appraisal of our study and the time taken to help improving the clarity of the manuscript. We have modified the title as suggested and the revised manuscript has been proofread. 

Formatting

Page numbers and line numbers are missing. The spacing between the references numbers is not appropriate. Please update the manuscript according to the guidelines of the journal.

We are sorry for the inconvenience for the reviewing process. In the revised manuscript, we have added page numbers and line numbers per page. It seems double-spacing was already used throughout the manuscript, including for the references.

Title

The authors did not estimate the burden of death statistics of HNSCC but highlight the under-reporting of death statistics. Therefore, I propose to revise the title of the manuscript such as “Factors associated with under-reporting of death statistics of HNSCC: A comparative study of two national databases in France from 2008 to 2012.”

We agree with the reviewer and the title has been changed accordingly into “Factors associated with under-reporting of head and neck squamous cell carcinoma in cause-of-death records: a comparative study of two national databases in France from 2008 to 2012”.

Abstract

Please add the following information:

• Objective: The authors can update and add another objective as follows: “Our objectives were to compare the reporting of death attributable to HNSCC from two national databases in France and to identify the underlying factors associated with under-reporting.” 

We agree with the reviewer and objectives have been modified accordingly in the abstract.

• Methods: Authors used the word “The determinants of missing HNSCC…, it is better to replace it with under-reporting.

We agree with the reviewer that it is better to use the same terminology and “determinants of missing HNSCC” has been replaced by “factors associated with under-reporting of HNSCC” throughout the revised manuscript.

• Results: Please revise the sentence structures for the results section. 

The revised manuscript has been proofread. 

• Conclusions: Please revise the first sentence according to the updated title.

This has been done in the revised manuscript.

Introduction

• In the first paragraph, last sentence “to our knowledge […]”, authors used exposure to tobacco and alcohol account for larger population attributable fractions of HNSCC but did not take into account these factors in their analysis. It is unclear to me mention it here. I propose to remove it and rephrase it. 

We agree with the reviewer and we removed this from the Introduction in the revised manuscript. The ideas are now taken up in the Discussion.

• Please replace the word “combining” with “linking” individual records […] at the beginning of the second paragraph.

This has been corrected in the revised manuscript.

• Please revise the text to improve the level of writing and avoid using words like instead, around, possibly.

We have rewritten the second paragraph to improve clarity in the revised manuscript that has been proofread.

Materials and methods

• Please add as a sub-heading the study design (part of the text from sub-section of the definition of HNSCC attributable deaths at hospital) and study population (inclusion and exclusion criteria from S2 table and cite S2 table under this section) just before the data sources. 

Following the reviewer’s suggestion, we have added a Study design section at the beginning of Material and Methods in the revised manuscript. In addition, inclusion and exclusion criteria (S2 Table), which are specific to the retrospective analysis of the national hospital discharge database have been transferred from the Results to the corresponding section in the Material and Methods (Identification of HNSCC-attributable deaths at hospital).

• Please also add the definition of “unlikely cause of death” at the end of the sub-section of the definition of HNSCC attributable deaths at hospital).

This has been added in the revised manuscript.

• Please revise the text of the last paragraph of the sub-section of “characteristics of HNSCC attributable deaths at hospital” and mention the “end of life characteristics” as 1. palliative care and 2. place of death (under this characteristic, authors can report deaths recorded at different places). This description should correspond to the table, which is not numbered (I suppose, table 4). 

This has been modified in the revised manuscript and “place of death” is now described in the Material and Methods, consistent with the terminology used in Table 3. All Tables were numbered in the submitted manuscript and there was no Table 4. 

• Please specify death at six age groups in the main text under the section of UCoD statistics for HNSCC. 

We have specified age groups in the revised manuscript.

• Please add the sub-section of study outcomes before the statistical analysis (the first paragraph of statistical analysis could be moved to the study outcome section)

This has been done in the revised manuscript.

• Please revise the text to improve the level of writing

The revised manuscript has been proofread. 

Results

1. The authors can start the result section by reporting the “General characteristics of deaths attributable to HNSCC according to two national databases in France” as table 1 rather than table 2. 

We would prefer to keep the initial order of presentation of the results (main text and Tables 1 and 2) in the revised manuscript. It seems more appropriate first to describe the HNSCC-attributable in-hospital deaths (Table 1) and then to compare these in-hospital deaths with those extracted from the national UCoD statistics (Table 2).

a. Please add the word “National” UCoD statistics in table 1 (currently table 2)

This has been added in the revised manuscript.

2. The second sub-section of results could be “Death attributable to HNSCC at hospital” as table 2. 

Please see above about reordering of the presentation of results.

a. Under this section, please also report the “unlikely death attributable to HNSCC” in the main text. 

This has been added in the revised manuscript.

b. Authors can report the “relapse in the follow-up overall from 2008 to 2010 from S3table to the main text and in table 2 as well.

This has been added in the revised manuscript.

c. Citation of S2 table is not appropriate in this section. 

This has been removed in the revised manuscript.

d. Please remove the word “definition” from the legend of table 2 (currently table 1) and start with “Deaths attributable […]”.

This has been done in the revised manuscript

e. In figure 1, please report the respective percentages of probable and possible causes of death attributable to HNSCC. Currently, it is unclear, for example in 2008, under in-hospital death, 70% corresponds to both causes of death (probable and possible)? Is this figure based on the data from S3 table? 

We agree with the reviewer that Figure 1, which presents annual detection rates of HNSCC-attributable in-hospital deaths identified in the national UCoD statistics could also be used to document the proportion of probable and possible in-hospital HNSCC deaths (presented in S3 Table). In line with the reviewer’s suggestion, we have clarified this in the revised manuscript. Figure 1 was removed from the first sub-section of the results (about probable/possible cause of death) to the second sub-section (about detection rates of HNSCC-attributable in-hospital deaths identified in the national UCoD statistics). Citation of Figure 1 in the first sub-section of results (about probable/possible cause of death) has been replaced by citation of S3 Table. 

3. The last paragraph of sub-section “Comparison of HNSCC […]”, please replace the word “multiple primary HNSCC” with “decedent”, which does not correspond to the table S4table.

This has been corrected in the revised manuscript.

a. Please update the title as follows: “Comparison of HNSCC […] with UCoD statistics”. 

This has been corrected in the revised manuscript.

4. The last sub-section of the results is unclear with long sentences. I propose to revise it and split into short sentences to make it more comprehensible. 

This sub-section has been shortened to improve clarity in the revised manuscript.

a. Under this sub-section, a table is reported and not numbered. I guess it should be table 4. Please add the table number and heading/legend of this table.

This was actually the second part (elasticities) of Table 3. The two parts (relative risks and elasticities) of Table 3 have been reconciled when reformatting the revised manuscript.

Discussion

• The authors should revise the overall text of the discussion section without repeating the same results, avoid long sentences, report strengths and limitations in a more clear text. 

We somewhat disagree on the first point of the reviewer as it seems better to elaborate the discussion starting from the main results. In addition, while we agree that Introduction/M&M/Results should be as concise as possible (see agreements with the previous comments of the reviewer), we do much appreciate having no words count limit in Plos One to put results into perspective. The revised manuscript has been proofread.

• This section could be started such as “The result of this study showed the underreporting of death attributable to HNSCC in national UCoD as compared to national hospital discharge database.” 

This has been done in the revised manuscript and the first paragraph of the Discussion has been rewritten.

• Avoid using words such as massive, unexpectedly, rather, indeed, altogether, etc.

The revised manuscript has been proofread. 

• Authors mentioned again the same phrase as reported in the introduction section “To our knowledge […]”, it is nuclear. Please explain the reason to repeat this sentence.

This has been removed in the revised manuscript.

Conclusions

Please update the conclusions section and do not report the results in this section. 

This has been done in the revised manuscript.

 

Reviewer #2: Ms Michelle Gourley

Overall the paper is well written and the analyses presented is clear and strongly support the conclusions reached.

Thank you very much for your positive appraisal of our study.

Some minor comments to improve the paper are outlined below:

- The paper doesn't make it clear whether deaths with an associated cause of death recorded as HNSCC have been looked at. Analyses of Australian deaths data suggests that HNSCC deaths would increase by around one-third (33%) if associated causes of death are included. Can this be looked at in the French mortality data? If not, or if no deaths are recorded with HNSCC as an associated cause, then suggest making this more explicit in the paper.

National UCoD statistics only provide summary data for any selected UCoD (by year, gender, age group, region of residence). For this reason, we had no access to individual MCoD (multiple causes of death) nor to the number of HNSCC that may be coded as an associated cause of death. Following the reviewer’s comment, we have explicitly clarified that we had no access to MCoD data in the revised manuscript and discussed this point among the limitations of our study.

- The statistics reported in the paper do not include a variance measure (ie confidence intervals). Is it possible to include these, or if not, provide a reason why these have not been included or are deemed not to be needed for the analyses presented.

We did not calculate confidence intervals for summary statistics presented in Table 1 and 2 (counts), Figure 1 (counts), and Figures 2 to 5 (ASMR) because the summary statistics are related to observations at the national level and it was not relevant to account for sampling error. Otherwise, 95% confidence intervals were presented for all coefficients estimated in the multivariate Poisson regression (Table 3). 

- Paper could benefit from a gramma check to fix minor grammatical errors eg in the abstract results 'associated to' should be 'associated with'

This has been corrected in the revised manuscript that has been also proofread.

---

## [Decision Letter · Decision Letter 1]

15 Dec 2020

PONE-D-20-25023R1

Factors associated with under-reporting of head and neck squamous cell carcinoma in cause-of-death records: a comparative study of two national databases in France from 2008 to 2012

PLOS ONE

Dear Dr. Schwarzinger,

Thank you for submitting your manuscript to PLOS ONE. After careful consideration, we feel that it has merit but does not fully meet PLOS ONE’s publication criteria as it currently stands. Therefore, we invite you to submit a revised version of the manuscript that addresses the points raised during the review process.

We look forward to receiving your revised manuscript.

Kind regards,

Brecht Devleesschauwer

Academic Editor

PLOS ONE

Additional Editor Comments (if provided):

The reviewers appreciated your efforts to address their comments, but raised some remaining smaller editorial issues. These can be addressed in a final, minor revision round.

Reviewers' comments:

Reviewer's Responses to Questions

**Comments to the Author**

1. If the authors have adequately addressed your comments raised in a previous round of review and you feel that this manuscript is now acceptable for publication, you may indicate that here to bypass the “Comments to the Author” section, enter your conflict of interest statement in the “Confidential to Editor” section, and submit your "Accept" recommendation.

Reviewer #1: All comments have been addressed

Reviewer #2: All comments have been addressed

2. Is the manuscript technically sound, and do the data support the conclusions?

Reviewer #1: Yes

Reviewer #2: Yes

3. Has the statistical analysis been performed appropriately and rigorously? 

Reviewer #1: Yes

Reviewer #2: Yes

4. Have the authors made all data underlying the findings in their manuscript fully available?

Reviewer #1: Yes

Reviewer #2: Yes

5. Is the manuscript presented in an intelligible fashion and written in standard English?

Reviewer #1: (No Response)

Reviewer #2: Yes

6. Review Comments to the Author

Reviewer #1: Reviewer’s comments to authors

Abstract

• Results: Please add the word “national” before UCoD records [..] in the first phrase.

Materials and methods

• May be revise the title of the sub-section as “characteristics of HNSCC attributable deaths in national hospital discharge database (PMSI)”.

• Please add one sentence to introduce PMSI database in this section and explicit the abbreviation (programme de médicalisation des systèmes d’information).

• It is better to use “national hospital discharge database (PMSI)” than only PMSI. In France, it is well known but in general using this abbreviation may be confusing. Therefore, I suggest adding, “national hospital discharge database and mention in bracket the PMSI”. Please update it throughout the manuscript.

Results

• Please revise the title of the section as suggested above in the method section as “Deaths attributable to HNSCC in national hospital discharge database (PMSI)”.

• Update the title of table 1 as well.

Discussion

This section needs some revisions and please check with an English native speaker for the proofreading of this section. Some suggestion to update the followings items:

• Please update the sentence as “This study showed significant under-reporting […]”

• First paragraph, last sentence: « […] national UCoD records was less than half of the

the PMSI database. »

• Second paragraph, last sentence, could split in two sentences as follows:

« This situation seems particularly critical for HNSCC-attributable deaths, since exposure to tobacco and alcohol are major risk factors for HNSCC. Moreover, multiple other causes of death that are declared in death certificates, may be responsible for that. »

• Third paragraph, in second sentence, please remove the word “consistently”.

• On page 27

o line 2-3, please remove the word “taken together” and start the sentence with “These findings […].

o Line 6, please remove “U.S.”.

o Line 8-9, update the sentence as follows: “HNSCC-attributable deaths in PMSI database were considered as the rule than exception […].

o Line 13, please remove the word “generally”.

o Iine 16-17, please remove the work “again” and “in general”.

Conclusions

• Please remove the word “in particular”.

Reviewer #2: The authors have adequately addressed my previous comments.

One additional comment to consider is to add to the discussion that some of the head and neck SCC deaths may be hidden/included in the group of ICD 10 codes C76-C80 Malignant neoplasms of ill-defined, secondary and unspecified site. This could be the case where the certifying doctor has not provided specific details about the site and type of cancer on the medical certificate of cause of death.

7. PLOS authors have the option to publish the peer review history of their article (what does this mean?). If published, this will include your full peer review and any attached files.

Reviewer #1: **Yes: **Romana Haneef

Reviewer #2: **Yes: **Ms Michelle Gourley

---

## [Author Response · Author response to Decision Letter 1]

15 Jan 2021

Comments to the Author

1. If the authors have adequately addressed your comments raised in a previous round of review and you feel that this manuscript is now acceptable for publication, you may indicate that here to bypass the “Comments to the Author” section, enter your conflict of interest statement in the “Confidential to Editor” section, and submit your "Accept" recommendation.

Reviewer #1: All comments have been addressed

Reviewer #2: All comments have been addressed

2. Is the manuscript technically sound, and do the data support the conclusions?

Reviewer #1: Yes

Reviewer #2: Yes

3. Has the statistical analysis been performed appropriately and rigorously?

Reviewer #1: Yes

Reviewer #2: Yes

4. Have the authors made all data underlying the findings in their manuscript fully available?

Reviewer #1: Yes

Reviewer #2: Yes

5. Is the manuscript presented in an intelligible fashion and written in standard English?

Reviewer #1: (No Response)

Reviewer #2: Yes

Reviewer #2 who is a native English speaker found that both manuscript and R1 were “presented in an intelligible fashion and written in standard English”. Following the first comments of reviewer #1, the revised version of the manuscript (R1) has been entirely proofread by a native English speaker. We thank reviewer #1 for her continued scrutiny and all her comments were taken into account, although we do not believe that re-asking that R2 is proofread by a native English speaker is worth the extra time and money at this stage. 

 

Reviewer #1: Reviewer’s comments to authors

Abstract

• Results: Please add the word “national” before UCoD records [..] in the first phrase.

This has been added in the R2 version of the manuscript.

Materials and methods

• May be revise the title of the sub-section as “characteristics of HNSCC attributable deaths in national hospital discharge database (PMSI)”.

This has been modified in the R2 version of the manuscript into “characteristics of HNSCC attributable deaths in the national hospital discharge database” as subtitles should be better kept short and reference to “PMSI” is provided at all occurrences in the text (see below).

• Please add one sentence to introduce PMSI database in this section and explicit the abbreviation (programme de médicalisation des systèmes d’information).

This was already done in the R1 version of the manuscript (lines 3-4 page 4).

• It is better to use “national hospital discharge database (PMSI)” than only PMSI. In France, it is well known but in general using this abbreviation may be confusing. Therefore, I suggest adding, “national hospital discharge database and mention in bracket the PMSI”. Please update it throughout the manuscript.

We agree with the reviewer. This has been modified in the R2 version of the manuscript except in subtitles and Table/Figure titles where the mention of “(PMSI)” has been removed (see above).

Results

• Please revise the title of the section as suggested above in the method section as “Deaths attributable to HNSCC in national hospital discharge database (PMSI)”.

This has been modified in the R2 version of the manuscript (see above).

• Update the title of table 1 as well.

This has been modified in the R2 version of the manuscript (see above).

Discussion

This section needs some revisions and please check with an English native speaker for the proofreading of this section. Some suggestion to update the followings items:

Please see above our overall comment.

• Please update the sentence as “This study showed significant under-reporting […]”

This has been modified in the R2 version of the manuscript.

• First paragraph, last sentence: « […] national UCoD records was less than half of the PMSI database. »

This has been modified in the R2 version of the manuscript.

• Second paragraph, last sentence, could split in two sentences as follows: « This situation seems particularly critical for HNSCC-attributable deaths, since exposure to tobacco and alcohol are major risk factors for HNSCC. Moreover, multiple other causes of death that are declared in death certificates, may be responsible for that. »

We agree with the reviewer. This has been clarified in the R2 version of the manuscript.

• Third paragraph, in second sentence, please remove the word “consistently”.

This has been removed in the R2 version of the manuscript.

• On page 27

o line 2-3, please remove the word “taken together” and start the sentence with “These findings […].

This has been removed in the R2 version of the manuscript.

o Line 6, please remove “U.S.”.

This has been removed in the R2 version of the manuscript.

o Line 8-9, update the sentence as follows: “HNSCC-attributable deaths in PMSI database were considered as the rule than exception […].

We disagree with the reviewer as this point was already discussed in the previous paragraph.

o Line 13, please remove the word “generally”.

This has been removed in the R2 version of the manuscript.

o Iine 16-17, please remove the work “again” and “in general”.

This has been removed in the R2 version of the manuscript.

Conclusions

• Please remove the word “in particular”.

We disagree with the reviewer as our results may apply to cancer deaths in general and HNSCC in particular.

 

Reviewer #2: The authors have adequately addressed my previous comments.

One additional comment to consider is to add to the discussion that some of the head and neck SCC deaths may be hidden/included in the group of ICD 10 codes C76-C80 Malignant neoplasms of ill-defined, secondary and unspecified site. This could be the case where the certifying doctor has not provided specific details about the site and type of cancer on the medical certificate of cause of death.

We agree with the reviewer. This was already discussed in the third paragraph: “We also found that any distant metastasis recorded after HNSCC diagnosis (47%) was associated with an increased risk of HNSCC being under-reported in UCoD records [13]” and fifth paragraph: “Since lung cancer was the most frequent second primary cancer (16.1%), and may arise from metastatic spread of HNSCC [2], our findings suggest that certifying physicians usually simplify the “chain of events leading directly to death” by reporting lung cancer as the UCoD rather than the primary HNSCC site”.

---

## [Editor Report · Decision Letter 2]

19 Jan 2021

Factors associated with under-reporting of head and neck squamous cell carcinoma in cause-of-death records: a comparative study of two national databases in France from 2008 to 2012

PONE-D-20-25023R2

Dear Dr. Schwarzinger,

We’re pleased to inform you that your manuscript has been judged scientifically suitable for publication and will be formally accepted for publication once it meets all outstanding technical requirements.

Kind regards,

Brecht Devleesschauwer

Academic Editor

PLOS ONE
---

## [Editor Report · Acceptance letter]

22 Jan 2021

PONE-D-20-25023R2 

Factors associated with under-reporting of head and neck squamous cell carcinoma in cause-of-death records: a comparative study of two national databases in France from 2008 to 2012 

Dear Dr. Schwarzinger:

I'm pleased to inform you that your manuscript has been deemed suitable for publication in PLOS ONE. Congratulations! Your manuscript is now with our production department. 

Kind regards, 

on behalf of

Prof. Dr. Brecht Devleesschauwer 

Academic Editor

PLOS ONE